# Decreased dihydroartemisinin-piperaquine protection against recurrent malaria associated with *Plasmodium falciparum plasmepsin 3* copy number variation in Africa

Leyre Pernaute-Lau[1,2,3], Mario Recker [4,5], Mamadou Tékété[6,7], Tais Nóbrega de Sousa [1,8], Aliou Traore[6], Bakary Fofana[6], Kassim Sanogo[6], Ulrika Morris [1], Juliana Inoue[5], Pedro E. Ferreira [9,10], Nouhoum Diallo [6], Jürgen Burhenne [11], Issaka Sagara [6], Alassane Dicko[6], Maria I. Veiga [9,10], Walter Haefeli[11], Anders Björkman[1,12], Abdoulaye A. Djimde [6], Steffen Borrmann[5,11,13,15] & José Pedro Gil [1,14,15] ✉

Dihydroartemisinin-piperaquine (DHA-PPQ) is being recommended in Africa for the management of uncomplicated *Plasmodium falciparum* malaria and for chemoprevention strategies, based on the ability of piperaquine to delay re-infections. Although therapeutic resistance to piperaquine has been linked to increased copy number in plasmepsin-coding parasite genes (*pfpm*), their effect on the duration of the post-treatment prophylactic period remains unclear. Here, we retrospectively analyzed data from a randomized clinical trial, where patients received either DHA-PPQ or artesunate-amodiaquine for recurrent malaria episodes over two years. We observed an increase in the relative risk of re-infection among patients receiving DHA-PPQ compared to artesunate-amodiaquine after the first malaria season. This was driven by shorter average times to reinfection and coincided with an increased frequency of infections comprising *pfpm3* multi-copy parasites. The decline in post-treatment protection of DHA-PPQ upon repeated use in a high transmission setting raises concerns for its wider use for chemopreventive strategies in Africa.

In 2023, infections with *Plasmodium falciparum* accounted for 96% of the estimated 263 million malaria cases worldwide, with more than 95% of the 597,000 *P. falciparum*-related deaths concentrated on the African continent, predominantly in children younger than 5 years[1]. In sub-Saharan African countries, sustained high transmission rates drive the frequent re-treatment of children due to reinfection events, during the slow process of acquiring disease-limiting immunity. Dihydroartemisinin-piperaquine (DHA-PPQ) has been introduced as a second-line and third-line option in the antimalarial drug policy of several African countries[1]. Its use is motivated by the long terminal half-life of piperaquine ($t_{1/2} = 20-30$ days)[2], which offers superior post-

treatment prophylactic efficacy, as compared with the mainstay artemether-lumefantrine[3]. The suppression of rapid re-infections following treatment and, thus, a lower annual frequency of malaria episodes, is expected to contribute to a reduction in overall malaria morbidity[4].

In Africa, DHA-PPQ has consistently shown high chemotherapeutic efficacy in clinical trials. However, in the SE Asian low transmission setting of Cambodia, high treatment failure rates arose in multiple foci upon nation-wide introduction of DHA-PPQ in 2012[5], leading to the withdrawal of this artemisinin-based combination therapy (ACT) by the local authorities in 2016[6]. DHA-PPQ treatment failures

have been linked to increased copy numbers of the *P. falciparum plasmepsin 2* (*pfpm2*) and *3* (*pfpm3*) genes, part of a dense cluster of four plasmepsin genes (*pfpm1-4*), in a ca. 25 kb spanning region at chromosome 14[7]. These genes code for members of the plasmepsin family of aspartic acid proteases[7–9], located in the food vacuole of the parasite, being all involved in the digestion of hemoglobin.

In African high transmission areas, slowly declining subtherapeutic concentrations of piperaquine after DHA-PPQ treatment provide ample opportunities for selecting re-infecting parasites less sensitive to piperaquine. This is of particular concern since *plasmepsin* gene copy number variation (CNV) have recently been shown to circulate in African parasite populations[10]. In addition, a future increase in DHA-PPQ usage in Africa is conceivable, should treatment failure rates with artemether-lumefantrine continue to increase on the continent[11], prompting a potential switch in first-line treatment policies[11]. Moreover, DHA-PPQ is currently being considered for several chemoprevention strategies, which would mandate regular administrations in target populations and hence, translate into increased selection pressure on parasite populations by trailing plasma concentrations of piperaquine after DHA-PPQ treatments[12–14].

Here, we present a retrospective clinical and molecular analysis of the dihydroartemisinin-piperaquine versus the artesunate-amodiaquine (ASAQ) arm, part of the multi-center WANECAM randomized controlled trial performed in Southern Mali. The study involved both active and passive follow-up for a total of two years per patient with repeated, directly observed treatment for each new malaria episode[3]. This unique design allowed us to investigate the durability of the post-treatment prophylactic efficacy of DHA-PPQ while assessing the parallel evolution of *pfpm2* and *pfpm3* copy number variation as putative piperaquine resistance factors.

## Results
### Cohort
Between January 16, 2012, and May 18, 2013, 225 and 224 patients with microscopically confirmed uncomplicated malaria were recruited upon informed consent, to receive DHA-PPQ and artesunate-amodiaquine, respectively. Baseline characteristics are listed in Supplementary Table 1. Median follow-up times were similar in both treatment groups (729 days, range 16–790 for DHA-PPQ and 729 days, range 120–791 for ASAQ, respectively). There were 593 re-treatment episodes in the DHA-PPQ arm and 837 in the artesunate-amodiaquine arm. Rescue treatment with quinine was administered in 18 *versus* 22 cases of early recurrent parasitemia before day 28, respectively ($P = 0.85$). Fourteen patients in the DHA-PPQ and 11 patients in the artesunate-amodiaquine arm did not complete the 2-year follow-up (Supplementary Fig 1). In one study, one drug-unrelated death occurred from an unknown cause in the DHA-PPQ arm and another in the ASAQ arm (Supplementary Table 5) after treatment for the initial episode. No other serious adverse events were reported.

### Rapid decline of piperaquine post-treatment protection efficacy
In agreement with the original WANECAM trial analysis[3], the consolidated *P. falciparum* malaria annual incidence rate calculated for the 2-year individual follow-up was lower for DHA-PPQ compared to artesunate-amodiaquine (1.37, 95% CI [1.26,1.49] vs. 1.92, 95% CI [1.79, 2.06]), due to piperaquine's extended prophylactic effect[15,16]. This, however, was predominantly driven by the marked difference in the reinfection incidence density (estimated incidence / person / day) between the two study arms during the first year of study only (2012-2013), with incidence densities converging over subsequent years (Fig. 1A). Accordingly, we observed an increase in the risk ratio (RR) of re-infections in the DHA-PPQ treatment arm relative to the artesunate-amodiaquine arm, increasing from RR = 0.58 (95% CI [0.35, 0.89]) over the first 6 months, to RR = 0.92 (95% CI [0.85, 1]) by month 12 (Fig. 1B);

from month 12, there was no statistical difference in the risk of re-infections between the two study arms (all 95% CI's overlap with 1).

In accordance to the observed increase in the relative reinfection risk over time, there was a statistically significant decrease in the average time to re-infection in the dihydroartemisinin-piperaquine arm between 2012 and 2014 ($P < 0.005$ ANOVA; 2012: mean 86 days, IQR [59, 107]; 2013: mean 72 days, IQR [51, 87]; 2014: mean 72 days, IQR [48, 76]) (Fig. 1C). Notably, this decrease in time-to-reinfection was observed against an overall decrease in malaria incidence over the trial period, possibly due to the intense case management of the target population (total annual incidence rate 2012: 2.59, 95% CI [2.41, 2.78], 2013: 1.46, 95% CI [1.36, 1.55], 2014: 1.13, 95% CI [1.05, 1.23]).

### Post-treatment selection of *plasmepsin* 3 gene copy number variation following DHA-PPQ treatment
Increased *pfpm2* copy number variation (copy number ≥1.5) was not identified in parasites from pre-treatment infections but was detected at low frequency among recurrent infections for the DHA-PPQ arm (baseline samples prior treatment 0/173 *versus* 6/439 in recurrent infections; Fisher's exact test, $P = 0.2$), mostly in infections with relatively short inter-episode periods (less than 70 days) (Supplementary Fig. 2). This data is in agreement with a number of previous reports concerning the role of *pfpm2* CNV in *P. falciparum* in vivo response to piperaquine[7–9].

Following previously described CNV categorization[8], only 6 infections were considered to be dominant for multiple copies of *pfpm3* (copy number ≥1.5).

We observed a statistically significant increase in the frequency of mixed infections carrying *pfpm3* multi-copy variants in the DHA-piperaquine arm, as assessed by normalized CNV scores over time (Fig. 2A and B; mean normalized CNV scores 2012: 1.01, 95% CI [0.99,1.03]; 2013: 1.01, 95% CI [0.99, 1.03]; 2014: 1.17, 95% CI [1.11, 1.22], $P < 0.001$), suggestive of positive selection of this mutation in the parasite population driven by piperaquine pressure. Importantly, this increase was not observed in the artesunate-amodiaquine comparison treatment arm (Fig. 2B).

Consistent with the proposed role of *pfpm3* CNV in mediating reduced sensitivity to piperaquine and our observed distribution of *pfpm2* CNV, early re-infecting parasites encountering trailing subtherapeutic piperaquine concentrations were found to be dominated by parasite isolates with increased average *pfpm3* copy numbers (Fig. 3), which again was not observed in the artesunate-amodiaquine arm.

*pfpm2* and *pfpm3* CNV events were not found to be present in the same infection. We also did not find evidence for the presence of the structural breakpoint associated to the *pfpm1-3* hybrid found in South East Asia[7,17].

No statistically significant signal was found for the positive selection by piperaquine of any of the tested *pfcrt* SNPs during the follow-up. However, *pfcrt* 76 T and 356 T mutant SNPs were independently associated with decreased times to re-infection compared to the wildtype infections [mean reinfection period (days): 155.4 *vs* 141.8; 153.5 *vs* 135.8, for *pfcrt* 76 and 356 SNPs, respectively] (Supplementary Fig. 3).

## Discussion
Here, we have analyzed a large clinical trial dataset with 1430 *P. falciparum* malaria episodes in 449 patients randomized to receive either DHA-PPQ or artesunate-amodiaquine for the first episode at recruitment, as well as for each subsequent *P. falciparum* recurrence over an extended follow-up of 2 years, in a high transmission setting in Africa. This trial markedly differs from the majority of antimalarial, single-episode, efficacy studies characterized by shorter active follow-ups, typically up to day 28[18], not designed to assess the post-treatment prophylactic efficacy over long periods.

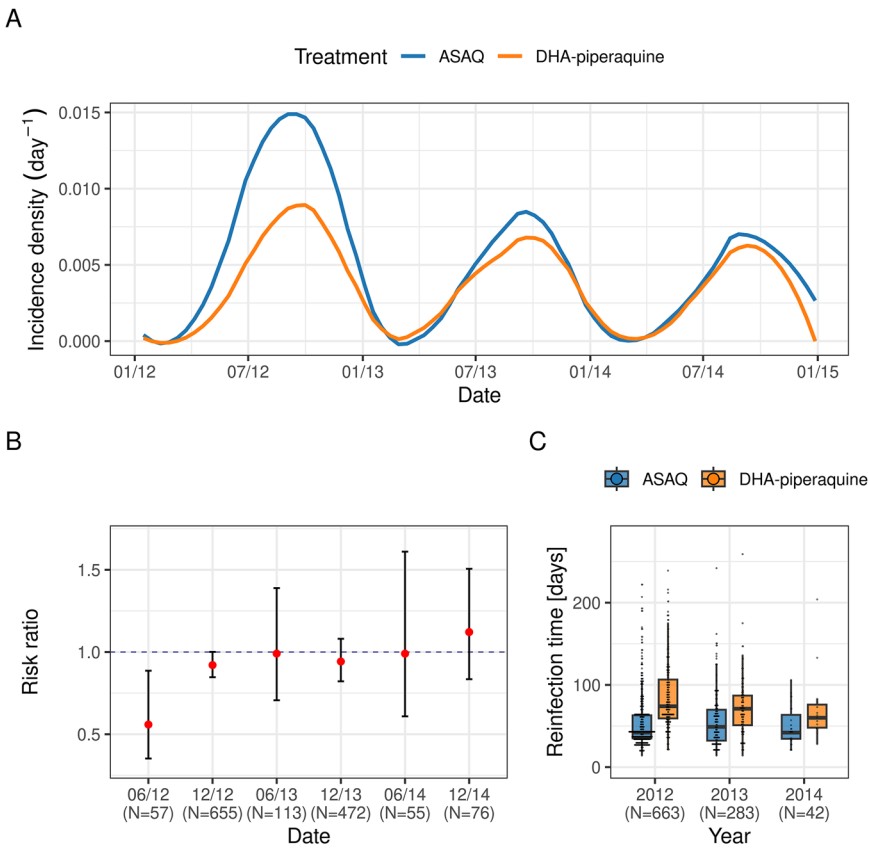

**Fig. 1 | Decline of long-term protection conferred by piperaquine. A** Estimated incidence density (number of infections / person / day) of clinical malaria episodes in the artesunate-amodiaquine (ASAQ, blue) and DHA-piperaquine (yellow) trial arms showing pronounced seasonality and decreasing transmission intensity in the study area together with a reduction in the long-term protective capacity of DHA-piperaquine, leading to converging incidence densities between the two arms over time. **B** Estimated risk ratios (DHA-piperaquine arm relative to the ASAQ arm, red circles) together with 95% confidence intervals demonstrating the decrease in long-term protection against reinfection over time with no statistically significant difference in (reinfection) risk from 06/2013 onwards. **C** Boxplot showing the decrease in the average time intervals between treatment and subsequent infection between 2012 and 2014 in the DHA-piperaquine arm ($P < 0.002$, ANOVA), but not the ASAQ control arm ($P = 0.9$, ANOVA).

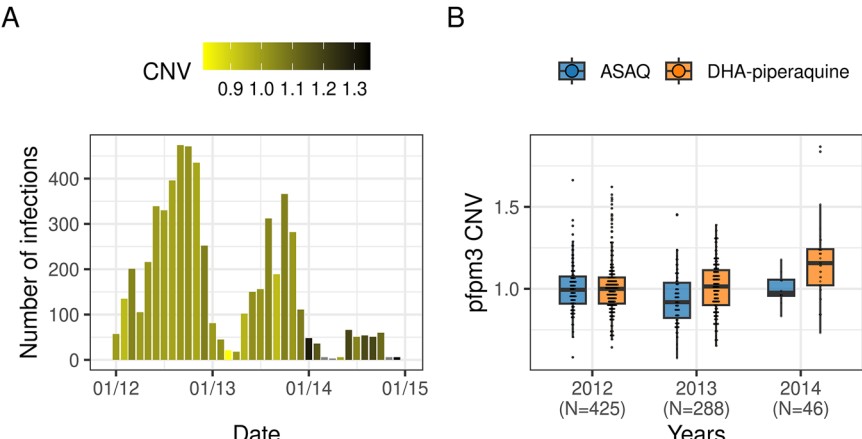

**Fig. 2 | Selection of multi-copy *pfpm3* parasites in the DHA-PPQ arm. A** Bar chart showing the number of clinical episodes in the DHA-PPQ trial arms, color-coded by average *pfpm3*-CNV, illustrating how selection appears to intensify along the high-transmission season (May-December) and towards the end of the trial, likely in response to an increase in drug pressure. **B** Boxplot of normalized *pfpm3* CNV demonstrating an increase of multi-copy variant infections between 2012 and 2014 in the DHA-piperaquine ($P < 0.001$, ANOVA); the overall trend in the ASAQ arm is slightly negative, driven by a temporary decline in 2013 ($P = 0.001$, ANOVA).

The extensive duration of surveillance allowed us to focus on the evolution of the post-treatment protective efficacy of DHA-PPQ over multiple years. Only two other trials with DHA-PPQ have reported results from repeated episodes[19,20]. This is an important knowledge gap since DHA-PPQ is being recommended and advocated on its capacity to provide longer protection against re-infections, thus reducing malaria morbidity compared to artemisinin-based combinations with less long-acting partner drugs.

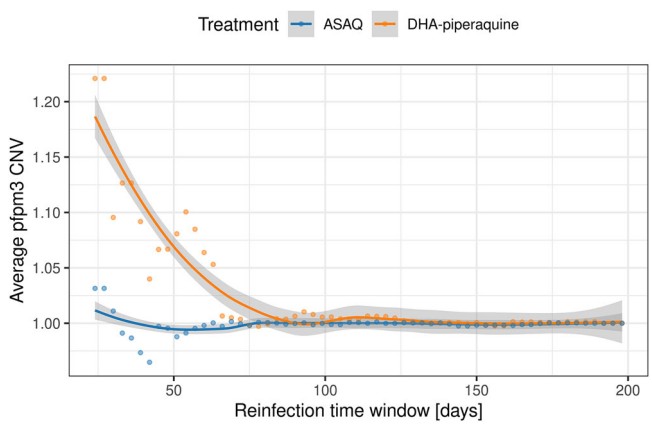

**Fig. 3 | *pfpm3* gene dose effect vs time to reinfection following DHA-piperaquine and ASAQ treatment.** Increased average *pfpm3* copy number is linked with a decrease in the time interval between treatment and reinfection, with early re-infections enriched with parasites carrying more than one *pfpm3* gene copy in the DHA-piperaquine arm (orange dots); this effect is not found in the ASAQ arm (blue dots). CNVs are scaled to aid comparison between both arms. Local regression lines (LOESS) are added to guide the eye; shaded areas indicate 95% confidence intervals.

We provide evidence that the post-treatment prophylactic efficacy of DHA-PPQ can wane rapidly. The observed decrease, within the first study year, of the key advantage of DHA-PPQ over an older and established ACT (ASAQ) provides a critical context for its proposed use for chemopreventive strategies, such as intermittent preventive treatment in pregnancy (IPTp)[13,21] or seasonal malaria chemoprevention in children (SMC)[12]. Of note, this finding required a time-dependent statistical approach since both the analysis of the multicenter trial data[3] and our initial analysis of the combined data from all years in Bougoula-Hameau concealed such critical changes over time.

Piperaquine monotherapy was widely used for both treatment and prophylaxis in China for more than a decade, until resistance developed in the late 1980s[22]. The drug was later reintroduced as part of several combinations, of which only the highly efficacious DHA-PPQ was retained[23]. However, the pharmacological mismatch between piperaquine and dihydroartemisinin ($t_{1/2} > 20$ days *vs* $t_{1/2} < 60$ min, respectively)[24] results in an extended, single drug pressure of piperaquine on re-infecting parasites mirroring the previous monotherapy used for mass drug administration applications (MDA) in China. Very few data are available on the long-term robustness of piperaquine's protective effect in these conditions. To our knowledge, the only multi-year study comprising a 5-year repeated treatment trial in Uganda showed a small reduction of malaria incidence in young children allocated to receive DHA-PPQ compared to artemether-lumefantrine[19]. However, as this analysis did not include the evolution of re-infection intervals over time, nor considered changes in the underlying parasite genotype distribution – the trial predated the discovery of the role of *plasmepsin* in piperaquine resistance – it is unclear how those results relate to the phenomenon reported here.

The reported decrease in protective efficacy in our study was specifically correlated with an increase in recurrent parasites carrying amplified *pfpm2 or 3*. Importantly, samples carrying *pfpm2 or 3* multi copies (considering a threshold of a ≥1.5 copies score) were not present at the baseline population, before the start of the treatment trial, suggesting that these have emerged upon piperaquine pressure. Consistent with this observation, samples from infections with short periods between treatment and subsequent episodes were found enriched in *pfpm2* and *pfmp3* CNV carrying parasites, suggesting an increased capacity of these parasites to re-invade under higher piperaquine concentrations. In addition, both outcomes were not identified in the artesunate-amodiaquine comparison treatment arm, reinforcing

the specificity of the association between the *pfpm2* and *pfmp3* molecular marker and the duration of post-treatment prophylaxis.

Notably, the chemotherapeutic efficacy of DHA-PPQ remained high throughout the duration of the trial, indicating that *pfpm2* and *pfmp3* copy number variation alone does not confer high-level resistance. Nevertheless, in scenarios with constant low drug pressure, fully resistant phenotypes can result from the accumulation of other mutations conferring an equilibrium between fitness cost and drug tolerability. In this context, it is to note that the *pfcrt* 76 T, 326S and 356 T SNPs were found enriched in shorter inter-episode infections. These might be contributors for this equilibrium during future resistance selection, in particular in the event of new emerging piperaquine resistance associated mutations in this gene[25].

Although partial artemisinin resistance was not reported in Africa at the time the trial was performed (2012–2015)[26], the current spread of *Kelch13* gene mutations in some countries[27–29] could potentiate the risk of treatment failures in case proper partner drug protection is not preserved.

Our results suggest that the impact of plasmepsin mutations on large-scale preventive interventions needs to be considered. Regular determination of *pfpm2* and *pfpm3* average CNV could be a potentially useful approach to monitor the evolution of parasite subpopulations under piperaquine pressure. This is specifically relevant in the current context of DHA-PPQ as an alternative to MDA in Africa, including protective measures like IPTp and SMC, which are expected to benefit from the long terminal elimination half-life of piperaquine[30].

Our findings raise an important question regarding the optimal use of this drug combination in national malaria control programs. Based on our results, there is a reasonable concern about the further evolution of *P. falciparum* resistance due to the continued large-scale exposure of the parasite to sub-therapeutic concentrations of piperaquine. Although the *pfpm2* and *pfpm3* CNV contribution for *P. falciparum* response to piperaquine in African high-transmission settings seems to be different from the reported in SEA, it could be exacerbated with emerging novel *pfcrt* mutations[31].

In conclusion, our study revealed that parasite populations appear to be able to rapidly respond to piperaquine pressure, leading to a decline in the long-term prophylactic advantage of DHA-PPQ in our study. Furthermore, genotyping of *pfpm2* and 3 CNV prevalence would be useful for the surveillance of the post-treatment prophylactic advantage of DHA-PPQ in African high transmission setting.

## Methods
### Patients and trial design
We retrospectively analyzed clinical, parasitological, and molecular data from all patients enrolled into the DHA-PPQ (DHA-PPQ) and artesunate-amodiaquine (ASAQ) arms in Bougoula-Hameau (southern Mali) of the West African Network for Clinical Trials of Antimalarial Drugs (WANECAM) clinical trial (PACTR201105000286876). The clinical trial was performed from 2011–2018 as a randomized, multicenter, open-label phase 3b/4 clinical trial at seven centers in Burkina Faso, Guinea and Mali. The DHA-PPQ and artesunate-amodiaquine arms in Bougoula were active from 2012 to 2015. Global primary and secondary endpoints have been previously reported[3]. The study was approved by local ethics committees and participating institutions (Ethical Committee of the Faculte de Medecin et D'Odonto-Stomatology, ref. 2010_79_FMPOS as well as by Stockholm Regional Ethics Committee, ref. 2016/2286-32 and 2017/499-32). Patients of either sex, aged ≥6 months with uncomplicated, microscopically confirmed *Plasmodium* spp malaria (axillary temperature ≥37.5 °C, or oral, or rectal, or tympanic temperature ≥38 °C or history of fever in the previous 24 h, and parasite density >0 to <200,000 parasites per μL blood) were randomly assigned to receive either DHA-PPQ (Alfasigma, Italy) or artesunate-amodiaquine (Sanofi, France) for each successive malaria episode during the 2-year follow-up of each patient (except for rescue treatment for episodes

within 26 days after start of treatment) (Supplementary Fig 1). Exclusion criteria included a hemoglobin concentration of < 7 g/dL, non-malarial febrile conditions, anti-malarial treatment within the previous two weeks, significant liver or renal impairment, and pregnancy. Once daily dosing for 3 days was per bodyweight (Supplementary Table 2). Patients were hospitalized during the directly observed treatment phase. After inclusion and after at each subsequent episode patients were followed up actively (weekly by microscopy) and after 63 days, passively by inviting study participants to present for any subsequent fever episode for a period of two years. As originally reported in the WANECAM clinical trial[3], recrudescence was distinguished from reinfection using *msp1, msp2*, and microsatellite markers. Briefly, for uncomplicated *P. falciparum* malaria, PCR-adjusted adequate clinical and parasitological response (ACPR) was greater than 99.5% at day 28 and greater than 98.6% at day 42 for all ACTs included.

## Molecular analysis of plasmepsin genes *pfpm2* and *pfpm3*

Genetic analysis was performed on samples collected at enrollment and upon parasite recurrence prior to re-treatment. DNA from dried blood spots on Whatman 3-mm filter paper (FP) was extracted with the QIAamp DNA Mini kit (Qiagen, Germany) according to the manufacturer´s instructions.

*Pfpm2* (PF3D7_1408000) gene copy number was assessed by a SYBR green-dye-based real-time polymerase-chain-reaction (PCR) assay in triplicates using a modified version of a previously described protocol[8,32] (Supplementary Table 3). Copy numbers of the *pfpm3* (PF3D7_1408100) gene were assessed by a TaqMan probe-based protocol adapted from Ansbro et al. (2020)[17] (Supplementary Table 3). The presence of the structural breakpoint of the *plasmepsin 1/3* amplification, previously identified in SE Asia parasites, was assessed in every infection showing evidence of carrying *plasmepsin 2* multiple copies[7,17]. The *P. falciparum* 3D7 strain was used as an external single gene copy control, the *β-tubulin* (PF3D7_1008700) gene served as an internal control, and a previously designed genetically modified clone with 2 copies of the *pfpm2* and *pfpm3* genes functioned as the control for multicopies[32].

PCR-Restriction Fragment Length Polymorphism (PCR-RFLP) and Sanger sequencing methods were used for the analysis of *pfcrt* SNPs (K76T, H97Y, C101F, F145I, N326S, and I356T), previously associated with increased piperaquine tolerability (Supplementary Table 4).

## Data analysis

The endpoints of our analyses were (i) the DHA-PPQ *versus* ASAQ relative risk of re-infection following antimalarial treatment in different study years, (ii) the time between treatment and re-infection, and (iii) the *pfpm3* gene CNV of the infecting parasite in patients allocated to DHA-PPQ. In addition, *pfpm3* gene copy number variation of artesunate-amodiaquine treatment arm infections was assessed as a comparison control arm.

Incidence density, or daily re-infection rates, were estimated using the total number of re-infection events over a 7-day period and normalized by the number of individuals in the respective trial arm, thus taking into account individuals entering and leaving the different treatment arms over time; data smoothing for visualization purposes was performed using R's local polynomial regression (LOESS) function *loess*.

Risk ratios, estimated using unconditional maximum likelihood estimations as encoded in the *riskratio()* function of the *epitools* R library (https://doi.org/10.32614/CRAN.package.epitools), were determined by comparing 6-months cumulative re-infection rates of DHA-PPQ treated individuals against those treated with ASAQ. For this analysis we only considered the period 2012–2014 due to the lack of samples from the DHA-piperaquine arm for 2015.

Due to the strong seasonality in malaria transmission in these settings, only infections that occurred within the same year between March and December were included in the calculation of the time to reinfection. Statistical significance between year-stratified mean reinfection times or mean *pfpm3* CNV was determined using Welch Two Sample t-tests or ANOVA, depending on the number of categories compared.

As the two study arms were analyzed with different batches of Taqman® probes for determining *pfpm3* CNV, and because the focus of this study was on their relative changes over time, and in comparison between the two study arms, all CNV scores were normalized against the respective average scores based on the first 50 infections in each arm, to permit direct comparison of temporal trends in this mutation over the study duration.

Re-infection time distributions are presented using box and whisker plots, where boxes indicate the median (bar), first and third quartiles (hinges), and the lower and upper whiskers indicate the smallest and largest value no further than 1.5 times the interquartile range from the hinges. All data analysis was performed using the R open-source statistical software, version 4.2 (R Project for Statistical Computing).

## Reporting summary

Further information on research design is available in the Nature Portfolio Reporting Summary linked to this article.

## Data availability

The datasets generated during and/or analyzed during the current study are available from the corresponding author upon reasonable request, with the agreement of all co-authors.

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

## Acknowledgements

This work was supported by a Swedish Research Council Grant (no. 2021-05666, ref. 2021-06048 and ref. 2021-03105) (J.P.G.). The WANE-CAM study was funded by the European and Developing Countries Clinical Trial Partnership and by the Medicines for Malaria Venture (Geneva, Switzerland) and is co-funded by the United Kingdom Medical Research Councils, the Swedish International Development Cooperation Agency, the German Ministry for Education and Research, the University Claude Bernard (Lyon, France), the University of Science, Techniques, and Technologies of Bamako (Bamako, Mali), the Center National de Recherche et de Formation sur le Paludisme (Burkina Faso), the Institut de Recherche en Sciences de la Santé (Bobo-Dioulasso, Burkina Faso), and the Center National de Formation et de Recherche en Santé Rurale (Guinea) (*all authors*). This work has been funded by Portuguese funds: Foundation for Science and Technology (FCT) - project UIDB/50026/2020, UIDP/50026/2020 and contract 2020.03113.CEE-CIND (M.I.V.). Projects NORTE-01-0145-FEDER-000039, supported by Norte Portugal Regional Operational Program (NORTE 2020), under the PORTUGAL 2020 Partnership Agreement, through the European Regional Development Fund (ERDF) (J.P.G.). Conselho Nacional de Desenvolvimento Científico e Tecnológico (CNPq), Brazil, Grant ref. 200075/2022-5 (T.N.S.). In addition, funded by the European and Developing Countries Clinical Trial Partnership (EDCTP), Medicines for Malaria Venture (MMV, Geneva, Switzerland), Federal Ministry of Education and Research (BMBF, Germany), and German Research Foundation (DFG), German Academic Exchange Service (DAAD) (*all authors*). A fellowship from BioSys PhD program PD65-2012 (Ref SFRH/BD/142860/2018) from Fundação para a Ciência e Tecnologia (Portugal) (L.P.L.).

## Author contributions

L.P.L. performed the experimental design, data analysis, data curation, and original draft writing. M.R. performed the data analysis, data curation, and visualization and contributed to writing. J.P.G. coordinated the conceptualization, design, draft writing, funding acquisition, and project administration. S.B. contributed to study design, data analysis, draft writing, and project administration. L.P.L., M.R., S.B., and J.P.G. contributed equally to data interpretation and writing (review and editing). M.T., T.N.S., U.M., and J.I. contributed to data acquisition. A.T., B.F., K.S., N.D., I.S., A.D., and A.A.D. were field-based collaborators responsible for the WANECAM Clinical trial. P.E.F., J.B., M.I.V., W.H., A.B., and T.N.S. contributed to writing (review and editing).

## Funding

## Competing interests

We declare no competing interests.

## Additional information

[1]Department of Microbiology and Tumour Cell Biology, Karolinska Institutet, Stockholm, Sweden. [2]BioISI – Biosystems & Integrative Sciences Institute, Faculty of Sciences, University of Lisbon, Lisbon, Portugal. [3]The Art of Discovery, Derio, Basque Country, Spain. [4]Centre for Ecology and Conservation, University of Exeter, Penryn Campus, Penryn, UK. [5]Institute for Tropical Medicine, University of Tübingen, Tübingen, Germany. [6]Malaria Research and Training Center, Faculty of Pharmacy, University of Science, Techniques and Technologies of Bamako, Bamako, Mali. [7]Dept. of Clinical Pharmacology, University of Heidelberg, Heidelberg, Germany. [8]Molecular Biology and Malaria Immunology Research Group, Instituto René Rachou, Fundação Oswaldo Cruz (FIOCRUZ); Belo, Horizonte, Brasil. [9]Life and Health Sciences Research Institute (ICVS), School of Medicine, University of Minho, Campus de Gualtar, Braga, Portugal. [10]ICVS/3B's—PT Government Associate Laboratory; 4806-909 Guimarães, Braga, Portugal. [11]German Center for Infection Research (DZIF), Tübingen, Germany. [12]Department of Global Public Health, Karolinska Institutet, Stockholm, Sweden. [13]Centre de Recherches Médicales de Lambaréné (CERMEL), Lambaréné, Gabon. [14]Clinical Tropical Medicine (CTM), Global Health and Tropical Medicine, Institute of Hygiene and Tropical Medicine, Nova University of Lisbon, Lisbon, Portugal. [15]These authors contributed equally: Steffen Borrmann, José Pedro Gil. ✉e-mail: jose.pedro.gil@ki.se

