## [Transparent Peer Review file · Nature Communications]

Decreased dihydroartemisinin-piperaquine protection against recurrent malaria associated with *Plasmodium falciparum* plasmepsin 3 copy number variation in Africa.

Corresponding Author: Professor Jose Pedro Gil

Version 0:

Reviewer comments:

Reviewer #1

(Remarks to the Author)

This is a report analyzing Plasmepsin copy number variation from a clinical trial evaluating repeated treatment with dihydroartemisinin-piperaquine (DHA-PPQ) vs. artesunate-amodiaquine (ASAQ) for each new episode of malaria over a two year period. The study addresses an important question about the evolution of resistance with repeated use of DHA-PPQ because of the long half-life of PPQ and supports the role of pfpm CNV monitoring for early detection of PPQ resistance. However, as I understand the data, it seems that the authors overstate their results.

Major concerns

1. The results are associative. Although the increased rate of reinfection compared to ASAQ suggests changes in DHA-PPQ efficacy over time, the association with time to next infection is not compared to ASAQ.
2. Median copy numbers never reach the stated value to define "multicopy" therefore the changes in median pfpm3 copy number might not be clinically significant.
3. Data are a decade old and its application to the current policy landscape is not clear.
4. Results: With over 800 person years of follow up, it seems unlikely (though possible) that there was only one serious adverse event. Please confirm this.
5. Results/Discussion: the authors point out that participants in this study were intensely followed and treated. Could this have accounted for changes in susceptibility to infection and disease? For example, children who receive chemoprophylaxis sometimes develop increased rates of infection/disease after discontinuation.
6. What was the distribution of pfpm3 CNV in the ASAQ population? Figure 2 Panel A and B: Need a comparison to ASAQ to determine if this is due to DHA-PPQ drug pressure. While I realize this is shown in Supp Figure 2, the results cannot be compared because they are depicted differently.
7. The data suggest that infection that occur while PPQ is present, but likely subtherapeutic, have a slightly higher CNV than infection that occur without drug pressure after ASAQ. But these infections have similar low CNV during most of the follow up period. This, it makes it difficult for me to understand how infections in single individuals would have more resistant infections over a two year period. In addition, is it correct that all reinfections beyond 28 days were treated with the same drug (eg all dots on Figure 3)? So did all increase CNV infections clear with DHA-PPQ?
8. Statement "pfpm2 and pfpm3 CNV events were not found to be linked"—where is this information?
9. Recent genetic crosses suggest that pfpm CNV does not contribute to resistance in the absence of a specif PfCRT SNP (367) that was not assessed in this study (Kane et al mBio July 2024).
10. The statement "the post-treatment prophylactic efficacy of dihydroartemisinin-piperaquine can wane rapidly" is not well-justified, as its efficacy remains higher than ASAQ throughout. In fact, subsequently, the following is stated (which is supported by the data) "the chemotherapeutic efficacy of dihydroartemisinin-piperaquine remained high throughout the duration of the trial."
11. Could the finding of higher CNV over the course of the rainy season reflect more complexity of infections with just an increased change of higher CNV? Also, what were the results for ASAQ? Is CNV associated with complexity of infection?
12. Figure 1: Are any of these differences statistically significant?
13. Supp Figure 1: I cannot follow this figure. If this could be made clearer, it may answer some of my questions.

Minor concerns

1. Introduction: " which offers superior post-treatment prophylactic efficacy" Superior to what?
2. Introduce acronym DHA-PPQ
3. No need for acronyms that are used infrequently such as DOT
4. Please explain what this means in results: "In agreement with the previously published global analysis...."
5. Please clarify if recrudescence infections were included or excluded from the analysis.
6. Figure 1 Please add a p-value comparing the incidence of malaria in DHA-PPQ vs ASAQ
7. Supp Table 1: What values are in parentheses? Why is 0 parasite density listed?

Reviewer #2

(Remarks to the Author)

This article looks at the prophylactic effect and selection pressure of DHA-PPQ in a high transmission setting in Mali. This data presented will be of interest to many given the increasing reports of reduced artemether-lumefantrine efficacy across Africa. This article, however, requires revision before it can be considered for publication. I have listed some points below to assist with the revision.

General:

1. If abbreviations are introduced, please use them in the text thereafter.
2. The results section contains a lot of discussion points which really should only appear in the discussion section. The results obtained from the current study should only be described in the results sections
3. Suggest the article is reviewed for flow and clarity before re-submission

Introduction:

1. Ln74: This statement is not true for malaria-endemic countries in Africa – please revise.
2. Ln77: Are you saying that DHA-PPQ has only been included at the policy level but has not been deployed? Please clarify.
3. Ln79: Do not understand the statement about preventing rapid re-infection. I thought the long half-life of the piperazine, meant longer protection against reinfection compared to artemether-lumefantrine for example. Please provide clarity.
4. Ln81-82: Provide a reference for the statement
5. Ln83: Please make it clear that the low transmission setting you are referring to is not from Africa.
6. Ln85: "swiftly" is a relative term – so please provide some dates to add some context. The year the drug was introduced would help.
7. Ln87: Do you mean they occur in close proximity on chromosome 14?
8. Ln89: Insert "are" before "involved"
9. Ln93: Do mean less sensitive to ACTs in general or specifically piperazine?
10. Ln95: Do not understand the point being made here. If there is resistance, it should be an issue there the drug is first or second line as the treatment will still fail. Please rephrase for clarity.
11. Ln101: Please expand on what analysis was done. Were you assessing acceptability, efficacy, or cost-effectiveness?
12. Ln103: I assume you mean patients were followed up for 2 years with DOT for each repeat infection, rather than only patients with repeat infections were followed up for 2 years? Please clarify.

Results:

1. Ln111: Please check if months need to be written out in full. Also, state that malaria was confirmed in these patients recruited to receive the specific treatment.
2. Ln114: Was the shortest time for follow-up in the DHA-PPQ arm 2 days? This person should have been excluded from the study given the short follow-up time.
3. Ln115: Was the difference in repeat infections significantly different? If possible, provide a p value here.
4. Ln122: Were there any minor adverse events? If so, they should be listed.
5. Ln125: This is a discussion point – just describe your result here, the findings can be explained in the discussion.
6. Ln128: Was the difference in incidence rate significant between the two treatments?
7. Ln136: Did the risk stay the same for ASAQ?
8. Ln137: Do not understand the point being made. Please clarify. Also state which drug you are talking about here.
9. Ln145: Why is only plasmepsin 3 mentioned in the title when you investigated both plasmepsin 2 and plasmepsin 3?
10. Ln154: Was the presence of 6 infections with increased plasmepsin 3 CNV significant? Were these repeat infections?
11. Ln158: Please quantify this increase over time – what was the number at baseline, after year one and at the end of the study.
12. Ln165: some quantification of the number of repeat samples with CNV needs to be provided in the text.
13. Ln167: Do not understand the point being made. Please rephrase for clarity.
14. Ln170: Are you saying the pfcr SNPs were equally present in both study arms or that no SNP was selected for in the DHA-PPQ arm? Please clarify.
15. Ln171: Was this association in re-infection time was present in both study arms or only the DHA-PPQ arm?

Discussion:

1. Ln180-181: This statement makes it sound like the patients got a different drug following the first infection and only if they returned were they treated with either DHA-PPQ or ASAQ. Please clarify.
2. Ln182: I thought more follow-ups are up to 28 days – please check.
3. Ln184: This is a moot point as most studies are not designed to assess prolonged prophylactic effects.
4. Ln187-188: Since drugs start off highly effective, with efficacy declining as resistance mutations are acquired, I am assuming you mean this study allowed you to investigate the evolution of resistance/reduced efficacy. Please clarify.

5. Ln191-192: I assume you mean provide longer protection against reinfection. Please clarify.
6. Ln195: Did you really show this? There were fewer repeat infections in the DHA-PPQ arm compared to the ASAQ arm. DHA-PPQ efficacy also remained high. Also not information on new vs recrudescence infections is presented – so are we to assume every repeat infection is a repeat infection?
7. Ln202: Do not understand what “complex combinations” means. Please clarify.
8. Ln218: Did they carry CNV in both the plasmepsin 2 and 3 genes?
9. Ln219-220: Do not understand the point being made. Please rephrase for clarity.
10. Ln224: I am assuming you mean reinfection infections with increased copy numbers were more likely to be detected follow DHA-PPQ treatment? Please clarify.
11. Ln239-240: Do not understand the point being made. Please rephrase for clarity.
12. Ln242: Why only plasmepsin 3?

Methods:

1. Ln274: Suggestion ethics approval numbers are reported here.
2. Ln276: So patients with any malaria species were enrolled into the study despite DHA-PPQ and ASAQ recommended for falciparum malaria? Please confirm.
3. Ln287-288: Please provide more information on how and what done during passive and active follow-up.
4. Ln290: Information on msp1 and 2 marking was not presented in this paper. So were all repeat infections new infections?
5. Ln310: provide a reference for the pfcr SNP marking.
6. Ln320: Why only plasmepsin 3?
7. Ln332: Do not understand why risk ratios could not be calculated for samples without filter paper samples. Please more information for clarity.
8. Ln333: Are you suggesting that there is no malaria transmission during the low season? Please provide a reference to support this statement.

Version 1:

Reviewer comments:

Reviewer #1

(Remarks to the Author)

Thank you for your response to the reviews and your helpful revision. I am still not clear which differences are or are not statistically significant because figures show trends without statistical testing results and results in the text generally do not show statistical significance. I think if these can be clarified, the manuscript has important information to inform surveillance for drug resistance moving forward.

REVIEWER COMMENTS

Reviewer #1 (Remarks to the Author)

Major concerns

1. The results are associative. Although the increased rate of reinfection compared to ASAQ suggests changes in DHA-PPQ efficacy over time, the association with time to next infection is not compared to ASAQ.

In fact, this comparison constitutes a critical aspect of our study. Figure 3, for instance, firmly demonstrates that the association between higher *pfpm2* or *3* copy number variation (CNV) scores and shortened reinfection times is only found in the DHA-piperaquine arm, not in the ASAQ arm, clearly indicating the specificity of this effect. We have revised all figures, which now all include a direct comparison between both arms to avoid any further potential misunderstandings.

2. Median copy numbers never reach the stated value to define “multicopy” therefore the changes in median *pfpm3* copy number might not be clinically significant.

Firstly, we would like to emphasise that throughout our report we never claimed the observed associations as related with clinical failure of the therapy. The high efficacy demonstrated – described in detail in the initial clinical report – attest to it [WANECAM, 2018]. What we propose is that exposing the parasite population to subtherapeutic levels of a long terminal half-life antimalarial leads to a progression towards drug resistance, by selecting less responsive parasites (albeit still inside the therapeutic window), as has been previously proposed for the impact of the *pfmdr1* N86 mutation on the time to re-infection after artemether-lumefantrine (Sisowath et al., 2005, Hastings and Ward, 2005, Malmberg et al., 2013). This phase of drug resistance development is essentially invisible if considered solely from the perspective of therapeutic efficacy in the management of acute malaria, but of clear importance for malaria control programmes. Molecular markers of this pre-resistance phase (“tolerance” [Hastings and Ward, 2005]) are in this context important for its monitoring, allowing timely public health decisions.

Returning to the clinical relevance of our observations, we should stress that clinical relevance goes beyond treatment efficacy; an increase of *pfpm2* or *3* duplication prevalence will decrease post-therapy protection, leading to more frequent and earlier symptomatic reinfections. This is of critical importance – the disease burden from uncomplicated *P. falciparum* malaria is determined by the frequency of malaria episodes/reinfections. Vice versa, the impact of antimalarial interventions is measured by their ability to reduce the frequency of malaria episodes (nicely illustrated by the well-described effects of the partially efficacious, recently WHO-authorized RTS,S/AS01 and R21/Matrix-M malaria vaccines). This is therefore an important medical issue, with a detrimental effect on haemoglobin concentrations as a general indicator of health. Moreover, our data has direct implications for the long-term future of mass drug administration (MDA) strategies using piperaquine-based combinations, where DHA-PPQ is given at fixed time intervals and expected to (a) eliminate any existing infections during administration of treatment and, importantly, (b) delay reinfections. This includes the WHO endorsed use in IPTp and SMC. In this regard, we would like to refer to the historical fast rise of piperaquine resistance in Southern China during the MDA efforts of the 1980s [Davis et al., 2005]

Concerning the assessment of *P. falciparum* blood stage infections harbouring multiple copies of a specific genomic locus in the context of African malaria infections, we would like to emphasise several aspects. Firstly, virtually all infections in high transmission areas (such as Southern Mali) are multi-clonal. In fact, even an oversimplistic cross of just two different clones can generate >30 of unique/independent F1 progeny (Vaughan et al. 2015). This massive intra-host diversity of *P. falciparum* clones poses an obvious challenge for assessing associations between genetic markers and relevant phenotypes since infections are likely to contain variable ratios of clones with wildtype and mutant genotypes. In this context, we would also like to recall studies conducted in SE Asia, where it was evident that the exposure to regular Artesunate-Mefloquine treatment led to intra-host selection of the subpopulations carrying *pfmdr1* duplications [Uhlemann et al., 2005].

This is ever more challenging in the case of a quantitative mutation analysis, as it happens with copy number variation (CNV). The presence of non-wild type parasites is easily diluted in the complexity of the infection,

leading to qPCR-determined CNV values below the 1.5 conventional threshold being categorised as ‘single-copy’ infections, when in reality simply determined that at least 50% of the infection is constituted by duplication carriers. We are dealing with infections characterised by intra-host parasite subpopulations, where a qPCR-determined CNV score as low as of 1.3 can still indicate the presence of a robust minority subpopulation of parasites carrying 2 copies, ready to be selected by drug pressure. In order to account for this large cryptic reservoir of *pfpm3* CNV carriers and avoiding the loss of valuable information, we do not define the infections in a binary mode of carriers vs. not carriers, but rather as the median CNV score among the study populations over time. We also wish to point that our analysis is focusing on *relative* changes from baseline over time and in comparison to a control group in a randomised controlled study design. This specific approach allowed us to pick up early evolutionary changes, i.e., *before* reaching conventional multi-copy status, in the composition of re-infecting *P. falciparum* populations that correlated with shortened time to microscopically detectable blood stage reinfections.

Davis TM, Hung TY, Sim IK, Karunajeewa HA, Ilett KF. Piperaquine: a resurgent antimalarial drug. *Drugs*. 2005;65(1):75-87. doi: 10.2165/00003495-200565010-00004. PMID: 15610051.

Hastings IM, Ward SA. Coartem (artemether-lumefantrine) in Africa: the beginning of the end? *J Infect Dis*. 2005 Oct 1;192(7):1303-4; author reply 1304-5. doi: 10.1086/432554. PMID: 16136476.

Malmberg M, Ferreira PE, Tarning J, Ursing J, Ngasala B, Björkman A, Mårtensson A, Gil JP. Plasmodium falciparum drug resistance phenotype as assessed by patient antimalarial drug levels and its association with pfmdr1 polymorphisms. *J Infect Dis*. 2013 Mar 1;207(5):842-7. doi: 10.1093/infdis/jis747. Epub 2012 Dec 5. PMID: 23225895; PMCID: PMC3563306.

Sisowath C, Strömberg J, Mårtensson A, Msellem M, Obondo C, Björkman A, Gil JP. In vivo selection of Plasmodium falciparum pfmdr1 86N coding alleles by artemether-lumefantrine (Coartem). *J Infect Dis*. 2005 Mar 15;191(6):1014-7. doi: 10.1086/427997. Epub 2005 Feb 8. PMID: 15717281.

Uhlemann AC, Ramharter M, Lell B, Kremsner PG, Krishna S. Amplification of Plasmodium falciparum multidrug resistance gene 1 in isolates from Gabon. *J Infect Dis*. 2005 Nov 15;192(10):1830-5. doi: 10.1086/497337. Epub 2005 Oct 7. PMID: 16235185.

Vaughan AM, Pinapati RS, Cheeseman IH, Camargo N, Fishbaugher M, Checkley LA, Nair S, Hutyra CA, Nosten FH, Anderson TJ, Ferdig MT, Kappe SH. Plasmodium falciparum genetic crosses in a humanized mouse model. *Nat Methods*. 2015 Jul;12(7):631-3. doi: 10.1038/nmeth.3432. Epub 2015 Jun 1. PMID: 26030447; PMCID: PMC4547688.

3. Data are a decade old and its application to the current policy landscape is not clear.

This is an important concern, and we therefore wish to explain in detail why we believe that our findings are relevant now and in the foreseeable future.

Although piperaquine has now been adopted by a number of national malaria control programs, its implementation has been limited, with artemether-lumefantrine, and in some countries artesunate-amodiaquine, being the most widely used first-line treatments by a wide margin. In 2022 [WHO, 2023], only Cameroon, Ghana and Nigeria (the latter only for confirmed cases) included DHA-PPQ as an antimalarial treatment policy [WHO, 2023]. Besides limited and circumstantial MDA studies, the drug has not yet been used at a programmatic level for this type of strategy in Africa. Additionally, the use of DHA piperaquine for short experimental periods has not exerted a sufficiently long pressure for yielding genetic changes in the parasite population, according to a recently meta-analysis [Moss et al., 2022]. For these reasons, we do not expect that the *pfpm2* and *pfpm3* duplication background has changed significantly across the continent during the last ten years. Additionally, the most likely scenario of change would be the presence of higher prevalence of *pfpm2* and *pfpm3* increased copy number due to DHA-PPQ use, which would make our data all the more relevant, as one could expect that the post-treatment protection conferred by piperaquine against rapid re-infections might be influenced by the increased baseline presence of these mutations. At a time when DHA-piperaquine is being considered for chemo-preventive strategies, including the recently adopted seasonal malaria chemoprevention (SMC) and post-discharge malaria chemoprevention [PDMC Saves Life Consortium, 2024], our data is relevant for the long-term performance of these novel strategies.

We would also like to stress the original scientific value of our combined epidemiological/molecular analysis. To start, we can demonstrate for the first-time in a unique multi-episode/re-treatment/multi-year cohort that

even seemingly small changes in the relative constitutions of mixed/multi-clonal *P. falciparum* infections occur under selective pressure and that those ‘clinically silent’ changes (i.e., not leading to treatment failure; see our comment above) can nonetheless result in a comparatively rapid decline of a key public health benefit of artemisinin combination therapies involving very slowly eliminated partner drugs, such as piperaquine (terminal $T_{1/2} > 30$ days [Tarning et al., 2005]), in our case this manifested as a substantial reduction in the time to patent blood stage re-infection. Also, and as referred to previously, the possibility of monitoring the development of drug resistance during its early stages – i.e., whilst the treatment is still efficacious for the management of disease episodes – is fundamental for timely surveillance. Finally, all this is achieved by an updated conceptual approach of considering the analysis of copy number as a non-categorical variable, better adapted to the complex reality of *P. falciparum* infections in Africa, with implications for studying other drug-resistance candidate genes with copy number variations.

Moss S, Mañko E, Krishna S, Campino S, Clark TG, Last A. How has mass drug administration with dihydroartemisinin-piperaquine impacted molecular markers of drug resistance? A systematic review. *Malar J.* 2022 Jun 11;21(1):186. doi: 10.1186/s12936-022-04181-y. PMID: 35690758; PMCID: PMC9188255.

Hill J; PDMC Saves Lives Consortium. Implementation of post-discharge malaria chemoprevention (PDMC) in Benin, Kenya, Malawi, and Uganda: stakeholder engagement meeting report. *Malar J.* 2024 Mar 27;23(1):89. doi: 10.1186/s12936-023-04810-0. PMID: 38539181; PMCID: PMC10976733.

Tarning J, Lindegårdh N, Annerberg A, Singtoroj T, Day NP, Ashton M, White NJ. Pitfalls in estimating piperaquine elimination. *Antimicrob Agents Chemother.* 2005 Dec;49(12):5127-8. doi: 10.1128/AAC.49.12.5127-5128.2005. PMID: 16304183; PMCID: PMC1315981

WHO, 2003 – World Malaria Report 2023

4. Results: With over 800 person years of follow up, it seems unlikely (though possible) that there was only one serious adverse event. Please confirm this.

We are grateful to this reviewer for raising this question. Upon carefully reviewing the original clinical trial database, we found 2 serious adverse events; both judged to be unrelated to the study medication: one sudden death in the dihydroartemisinin-piperaquine arm; one case of meningitis in the artesunate-amodiaquine arm. We have updated this information in the results section (page 6, lines 129-132).

5. Results/Discussion: the authors point out that participants in this study were intensely followed and treated. Could this have accounted for changes in susceptibility to infection and disease? For example, children who receive chemoprophylaxis sometimes develop increased rates of infection/disease after discontinuation.

We believe that the reviewer might refer to an often hypothesised, but difficult to prove/show, ‘rebound’ effect. If so, we do not think that this could apply here, as this study did not involve conventional chemoprophylaxis (requiring dosing with fixed time intervals) and the intervention was not stopped at any time during the study period. Administration of DHA-PPQ or ASAQ was restricted to the management of each malaria episode for a duration of 2 years per study participant, according to the initial (episode #1) randomised allocation to either the DHA-PPQ or the ASAQ arm of the study.

We also wish to highlight two other aspects of our analysis. Firstly, and as mentioned above, the clinically relevant loss of the post-treatment protection against rapid re-infection only occurred in the DHA-PPQ arm of the study, but not in the parallel ASAQ arm, as illustrated in Figure 3. Secondly, we note that this was a regulatory level, GCP/ICH-compliant clinical trial with a strictly controlled randomised study design, which allowed us to separate global trends (such as the one referred to by this reviewer) from intervention specific-changes. Secondly, we observed a DHA-PPQ arm-specific reduction in the time to re-infection despite an overall reduction (both arms together) in the malaria transmission intensity.

6. What was the distribution of *pfmp3* CNV in the ASAQ population? Figure 2 Panel A and B: Need a comparison to ASAQ to determine if this is due to DHA-PPQ drug pressure. While I realize this is shown in Supp Figure 2, the results cannot be compared because they are depicted differently.

We thank the reviewer for bringing up this important point. For better clarity, we have revised the figures to always include a direct comparison between both arms. Furthermore, we have now normalised all CNV data against the respective average CNV of the first 50 infections in both study arms to facilitate better comparison in the temporal trend in CNV over the study period. We have updated Figure 2B to now show CNV distribution for both study arms between 2012 and 2014.

7. The data suggest that infection that occur while PPQ is present, but likely subtherapeutic, have a slightly higher CNV than infection that occur without drug pressure after ASAQ. But these infections have similar low CNV during most of the follow up period. This, it makes it difficult for me to understand how infections in single individuals would have more resistant infections over a two-year period. In addition, is it correct that all reinfections beyond 28 days were treated with the same drug (e.g. all dots on Figure 3)? So did all increase CNV infections clear with DHA-PPQ?

The same drug was used in the same patient in repeated episodes, except in instances where a recurrency was observed by D14 post treatment initiation, when oral quinine would be used as rescue treatment. Taking a previous analysis of the impact of short periods of intense DHA-PPQ [Moss et al., 2022] into consideration, we do not expect that the baseline population of parasites in the targeted region (Bougoula-Sikasso) changed dramatically as a result of this trial. In fact, in this sense the ASAQ arm serves as a control of any potential changes occurring to the overall population of parasites not exposed to DHA-PPQ. We believe that the repeated treatment with DHA-PPQ created a micro-environment of piperazine-driven enrichment and subsequent selection of multiclonal infections with higher average *pfpm3* CNV in this group of individuals. The very long termination half-life of piperazine (>30 days) leads to the remaining, low nM levels of drug in the body for more than 6 months post-treatment [Tärning et al., 2005]. In other words, in most patients there was a continuous presence of this drug during the 2-year follow-up. Considering that many of the patients experienced a recurrent infection before the end of the window of drug exposure, essentially all re-infecting parasites could be considered to be under continuous piperazine drug pressure, leading to selection for infections harbouring higher proportions of minority clones with increased *pfpm2* and *pfpm3* copy numbers.

8. Statement “*pfpm2* and *pfpm3* CNV events were not found to be linked”—where is this information?

We agree that this information is not fully described in the text. The statement was put forward in the context of PPQ resistance in SE Asia [Amato et al., 2017], which is associated with a specific rearrangement of plasmepsin genes, which are compacted in a head to tail arrangement in one region of chromosome 14. In these circumstances, a *pfpm1/3* hybrid is generated that has been interpreted as being a key event for the generation of resistance. We have tested this possibility through an inverted PCR approach, referred at page 8, lines 177-179. We did not detect the presence of this PPQ resistance related hybrid structure. We conclude that in the infections herein analysed, the increased in copy number represents the amplification of the full open reading frames. Additionally, we did not find convincing evidence of the presence of duplications in both genes in the same infection.

The text was modified for better clarity.

9. Recent genetic crosses suggest that *pfpm* CNV does not contribute to resistance in the absence of a specific PfCRT SNP (367) that was not assessed in this study (Kane et al mBio July 2024).

The G367C mutation identified as involved in DHA-PPQ resistance is of SE Asian origin. In the mentioned genetic cross, it is therefore introduced by the Cambodian KH004 artemisinin resistant clone. To our knowledge, this mutation has not yet been identified in Africa. We wish to emphasise again that our report pertains to a decreased post-treatment prophylactic effect, which, despite its specific and substantial public health importance, does not necessarily implicate drug resistance (in the sense of treatment failure). The effect of *pfpm* CNV in the absence of critical *pfert* mutations is likely to be small, but as pointed out critical for the duration of the post-treatment protection against rapid re-infections.

We would further like to note that we have analysed the proximal I356T SNP. Taking this codon position into consideration [Kim et al., 2019], as well as the importance of the South American PPQ-resistance associated

C350R variant, we hypothesize that the 356T allele may be implicated in the interaction of piperazine with pfCRT. Accordingly, we observed a significant trend of the 356T allele (previously associated to 4-aminoquinoline) to be associated with earlier recurrence in the DHA-PPQ arm, albeit independent of the *pfpm2* (none of the infections with increased copy number also carried *pfprt* 356T), neither *pfpm3*, where median number of copies not significantly different between the infections carrying the I356 vs 356T allele (Man-Whitney U, $p > 0.1$).

Kim J, Tan YZ, Wicht KJ, Erramilli SK, Dhingra SK, Okombo J, Vendome J, Hagenah LM, Giacometti SI, Warren AL, Nosol K, Roepe PD, Potter CS, Carragher B, Kosiakoff AA, Quick M, Fidock DA, Mancina F. Structure and drug resistance of the Plasmodium falciparum transporter PfCRT. Nature. 2019 Dec;576(7786):315-320. doi: 10.1038/s41586-019-1795-x. Epub 2019 Nov 27. PMID: 31776516; PMCID: PMC6911266.

10. The statement “the post-treatment prophylactic efficacy of dihydroartemisinin-piperazine can wane rapidly” is not well-justified, as its efficacy remains higher than ASAQ throughout. In fact, subsequently, the following is stated (which is supported by the data) “the chemotherapeutic efficacy of dihydroartemisinin-piperazine remained high throughout the duration of the trial”

As referred above, we would like to stress that a decrease in the protective capacity of piperazine during follow-up does *not* equate to nor indicate loss of adequate therapeutic efficacy. Therapeutic efficacy is capturing the *combined* effect on parasite elimination of both dihydroartemisinin and piperazine at peak drug concentration (i.e., during/shortly after dosing), which is still indeed intact, as reported before in the original paper (WANECAM, 2018). Here, we are reporting on the waning long-term protection of DHA-PPQ against rapid re-infections. This post-treatment prophylactic efficacy is governed solely by the slowly eliminated piperazine component of the combination (dihydroartemisinin is eliminated within a few hours after dosing). Since this involves substantially lower drug concentrations during follow-up, i.e., low concentrations that are sufficient to suppress wildtype parasite replication but unable to stop the replication of reinfecting parasites with even small decreases in the chemosensitivity. These small changes in the chemosensitivity of parasites, however, are obviously not able to confer resistance to much higher concentrations during treatment (and again of both DHA and PPQ).

As shown in Figure 1, the comparative advantage of DHA-PPQ over ASAQ for delaying reinfections decreased significantly after the first year, from an initial relative risk ratio of 0.58 (95% CI; 0.35, 0.89) increasing to 0.92 (95% CI; 0.85, 1) after 12 months only indicating a sharp loss of the initial advantage over ASAQ in delaying secondary reinfections.

Therefore, what we observed and report here was a rapidly waning post-treatment efficacy despite a stable therapeutic efficacy of DHA-PPQ, i.e., a dissociation of these two independently important effects of antimalarial treatment in the typically high-transmission areas of sub-Saharan Africa. Again, this is due to the much higher drug concentrations during treatment course as compared with the trailing levels during the long follow-up, which can fail to suppress the replication of parasites with even only slightly reduced drug susceptibility – hence leading to shorter time to reinfection as observed in our study.

11. Could the finding of higher CNV over the course of the rainy season reflect more complexity of infections with just an increased change of higher CNV? Also, what were the results for ASAQ? Is CNV associated with complexity of infection?

We cannot easily think of a mechanism whereby an increase in MOI during the rainy season would by itself causing an increase in the prevalence of *pfpm2* and 3 CNV. If anything, it is expected that such mutations might incur a fitness cost, which will handicap mutant parasites when facing the competition of wildtype parasites within multi-clonal infections (intra-host competition) [Fröberg et al., 2013]. We believe that the reason for these patterns is essentially due to the much higher frequency of treatment during the rainy season, and thus, higher drug pressure, which would favour parasites carrying a mutation that allows earlier reinfection [Dhingra et al., 2019]. Such effects have been observed also in other locations, where the malaria seasons are even more marked, an example being Southern Sudan, where it was consistently observed that the prevalence of chloroquine resistance associated mutations (*pfprt* 76T and *pfmdr1* 86Y) reached their peak by the start of the dry season, after the significantly increased treatment pressure during the preceding wet season. Without this pressure, the prevalence of wild type/drug sensitive parasites with higher average fitness will increase

among the mostly asymptomatic infections during the low malaria season [Abdel-Muhsin et al., 2004, Babiker et al., 2005]. Unfortunately, the information on the MOI is limited for this trial, as the biodiversity marker genotyping (*pfmsp1*, 2 and *glurp*) was focused on the analysis of recurrent infections and their baseline infections for PCR correction of efficacy.

Abdel-Muhsin AM, Mackinnon MJ, Ali E, Nassir el-KA, Suleiman S, Ahmed S, Walliker D, Babiker HA. Evolution of drug-resistance genes in *Plasmodium falciparum* in an area of seasonal malaria transmission in Eastern Sudan. *J Infect Dis.* 2004 Apr 1;189(7):1239-44. doi: 10.1086/382509. Epub 2004 Mar 16. PMID: 15031793.

Babiker HA, Satti G, Ferguson H, Bayoumi R, Walliker D. Drug resistant *Plasmodium falciparum* in an area of seasonal transmission. *Acta Trop.* 2005 Jun;94(3):260-8. doi: 10.1016/j.actatropica.2005.04.007. Epub 2005 Apr 25. PMID: 15857801.

Dhingra SK, Gabryszewski SJ, Small-Saunders JL, Yeo T, Henrich PP, Mok S, Fidock DA. Global Spread of Mutant PfCRT and Its Pleiotropic Impact on *Plasmodium falciparum* Multidrug Resistance and Fitness. *mBio.* 2019 Apr 30;10(2):e02731-18. doi: 10.1128/mBio.02731-18. PMID: 31040246; PMCID: PMC6495381.

Fröberg G, Ferreira PE, Mårtensson A, Ali A, Björkman A, Gil JP. Assessing the cost-benefit effect of a *Plasmodium falciparum* drug resistance mutation on parasite growth in vitro. *Antimicrob Agents Chemother.* 2013 Feb;57(2):887-92. doi: 10.1128/AAC.00950-12. Epub 2012 Dec 3. PMID: 23208719; PMCID: PMC3553735.

12. Figure 1: Are any of these differences statistically significant?

For the reinfection risk ratios (Fig1B), the 95% confidence intervals do not overlap with 1 (1 indicates no difference) until early 2013, which means that there was a statistically significant difference in risk between the two treatment arms before that date, which disappeared thereafter (i.e., there was no longer any difference in relative risk). The difference in the average time to reinfection depicted in Fig. 1C is statistically significant for the DHA-piperaquine arm between 2012 and 2013 ($P < 0.002$); we cannot make a statement for 2014 due to the sparsity of datapoints for that time period (see figure – start and final endpoint for the analysed set of patients). There was no statistically significant difference in the ASAQ arm. For Fig. 1A, please note that this is a continuous variable and not a summary statistic that can be compared using null-hypothesis testing.

13. Supp Figure 1: I cannot follow this figure. If this could be made clearer, it may answer some of my questions.

Supplementary Figure has been simplified for clarity, while additional information was added to the legend:

Upper panel represents the timeline distribution of *Plasmodium falciparum* *pfpm2* copy number status during DHA-PPQ treatment follow-up. Each circle represents a recurrent malaria infection; empty circles indicate *pfpm2* single copy infections and red circles indicate *pfpm2* multicopy infections (copy number ≥ 1.5). The x axis represents the number of follow up days until the next infection. *Pfpm2* multicopy infections present shorter inter-episode times than *pfpm2* single copy infections.

Minor concerns

1. Introduction: ” which offers superior post-treatment prophylactic efficacy” Superior to what?

Superior to the present mainstay African first-line ACTs, artemether-lumefantrine. The text was modified accordingly:

“Its use is motivated by the long terminal half-life of piperaquine ($t_{1/2} = 20\text{-}30$ days) ², which offers superior post-treatment prophylactic efficacy, as compared with the mainstay artemether-lumefantrine ³”

2. Introduce acronym DHA-PPQ

This is now introduced on page 4, line 79.

3. No need for acronyms that are used infrequently such as DOT

The text has been modified accordingly.

4. Please explain what this means in results: “In agreement with the previously published global analysis...”

We are referring to the extensive analysis performed in the original clinical trial report, which included the appraisal of all the trial arms. This complete analysis is what we refer as “global”. We understand now that the wording can induce the idea of a global as “worldwide” analysis. The text was changed to avoid this misunderstanding: “In agreement with the original WANECAM trial analysis...”

5. Please clarify if recrudescence infections were included or excluded from the analysis.

Confirmed recrudescence infections were rare in this study, reflected in the high efficacies of DHA-PPQ, >99% at D28, and >98% at D42 [WANECAM, 2018], and not excluded from the analyses as they are unlikely to have a significant influence on the results herein reported, especially as these also face the same drug-induced selection pressure (i.e. in the uncommon case of clearance failure, CNV carrying parasites are presumably more likely to recrudescence than wild-type parasites).

West African Network for Clinical Trials of Antimalarial Drugs (WANECAM). Pyronaridine-artesunate or dihydroartemisinin-piperaquine versus current first-line therapies for repeated treatment of uncomplicated malaria: a randomised, multicentre, open-label, longitudinal, controlled, phase 3b/4 trial. *Lancet*. 2018 Apr 7;391(10128):1378-1390.

6. Figure 1 – Please add a p-value comparing the incidence of malaria in DHA-PPQ vs ASAQ

As mentioned earlier, this is a continuous variable and not a summary statistic; it is not amenable to standard hypothesis testing, and we make no claim in that regard.

7. Supp Table 1: What values are in parentheses? Why is 0 parasite density listed?

Supplementary Table 1 has been modified to clarify the values in parentheses.

Reviewer #2 (Remarks to the Author)

General

1. If abbreviations are introduced, please use them in the text thereafter.

The manuscript has been fully revised and changed accordingly

2. The results section contains a lot of discussion points which really should only appear in the discussion section. The results obtained from the current study should only be described in the results sections

Although we understand the reviewer's perspective, we consider that the approach taken by us provides important reference points that put results into context and facilitates their in-depth interpretation in the discussion section. The journal guidelines do not mention provisions for avoiding the use of referencing previous work in the Results section. We therefore prefer the current format but await further editorial guidance.

3. Suggest the article is reviewed for flow and clarity before re-submission

In this revised the manuscript we also focused on improved clarity and readability of our manuscript. We hope that the editor and the reviewers will agree.

Introduction:

1. Ln74: This statement is not true for malaria-endemic countries in Africa – please revise.

In many sub-Saharan African countries, *Anopheles* mosquito-borne transmission of *P. falciparum* is intense, leading to frequent episodes of malaria. Impregnated bed nets are credited with a large proportion of the reduction of the disease burden from malaria (PMID 26375008). Likewise, the two recently WHO approved malaria vaccines RTS,S/AS01 and R21/Matrix-M can further reduce the frequency of malaria episodes (PMID 25913272 and PMID 38310910). Infections, however, cannot fully be prevented and thus, malaria control programmes are still relying on the long-standing WHO recommended 'early diagnosis and prompt treatment' policy to prevent life-threatening manifestations. Because acquisition of partial immunity that limits disease manifestations is slow and takes many years, children are the largest vulnerable population in need of frequent retreatment [World Malaria Report 2023].

We understand that this overall scenario was not sufficiently clearly stated and have thus modified the text to improve clarity.

Bhatt S, Weiss DJ, Cameron E, Bisanzio D, Mappin B, Dalrymple U, Battle K, Moyes CL, Henry A, Eckhoff PA, Wenger EA, Briët O, Penny MA, Smith TA, Bennett A, Yukich J, Eisele TP, Griffin JT, Fergus CA, Lynch M, Lindgren F, Cohen JM, Murray CLJ, Smith DL, Hay SI, Cibulskis RE, Gething PW. The effect of malaria control on *Plasmodium falciparum* in Africa between 2000 and 2015. *Nature*. 2015 Oct 8;526(7572):207-211. doi: 10.1038/nature15535. Epub 2015 Sep 16. PMID: 26375008; PMCID: PMC4820050.

RTS,S Clinical Trials Partnership. Efficacy and safety of RTS,S/AS01 malaria vaccine with or without a booster dose in infants and children in Africa: final results of a phase 3, individually randomised, controlled trial. *Lancet*. 2015 Jul 4;386(9988):31-45. doi: 10.1016/S0140-6736(15)60721-8. Epub 2015 Apr 23. Erratum in: *Lancet*. 2015 Jul 4;386(9988):30. doi: 10.1016/S0140-6736(15)60643-2. PMID: 25913272; PMCID: PMC5626001.

Datoo MS, Dicko A, Tinto H, Ouédraogo JB, Hamaluba M, Olotu A, Beaumont E, Ramos Lopez F, Natama HM, Weston S, Chemba M, Compaore YD, Issiaka D, Salou D, Some AM, Omenda S, Lawrie A, Bejon P, Rao H, Chandramohan D, Roberts R, Bharati S, Stockdale L, Gairola S, Greenwood BM, Ewer KJ, Bradley J, Kulkarni PS, Shaligram U, Hill AVS; R21/Matrix-M Phase 3 Trial

Group. Safety and efficacy of malaria vaccine candidate R21/Matrix-M in African children: a multicentre, double-blind, randomised, phase 3 trial. *Lancet*. 2024 Feb 10;403(10426):533-544. doi: 10.1016/S0140-6736(23)02511-4. Epub 2024 Feb 1. PMID: 38310910.

2. Ln77: Are you saying that DHA-PPQ has only been included at the policy level but has not been deployed? Please clarify.

As mentioned in our response to reviewer #1, although DHA-PPQ has been adopted by African countries, its deployment has been limited, especially considering the main ACTs, AL and ASAQ. The WHO 2023 World Malaria Report that compiles national treatment policies lists DHA-PPQ only as second or third-line therapeutic options in Cameroon, Ghana and Nigeria [World Malaria Report 2023].

3. Ln79: Do not understand the statement about preventing rapid re-infection. I thought the long half-life of the piperazine, meant longer protection against reinfection compared to artemether-lumefantrine for example. Please provide clarity.

That is correct, the extended post-treatment prophylactic window, and thus prevention of rapid reinfection, is a result of the long terminal half-life of PPQ, as stated in the text. This well documented capacity of PPQ performance is also the basis for the recent support of its use for post-discharge malaria chemotherapy (PDMC)[Hill et al., 2024][Phiri et al., 2024]. What we observe is the waning of this advantage over other artemisinin-based combinations following long-term exposure to piperazine upon multiple sequential treatments.

Hill J; PDMC Saves Lives Consortium. Implementation of post-discharge malaria chemoprevention (PDMC) in Benin, Kenya, Malawi, and Uganda: stakeholder engagement meeting report. *Malar J*. 2024 Mar 27;23(1):89. doi: 10.1186/s12936-023-04810-0. PMID: 38539181; PMCID: PMC10976733.

Phiri KS, Khairallah C, Kwambai TK, Bojang K, Dhabangi A, Opoka R, Idro R, Stepniewska K, van Hensbroek MB, John CC, Roberstad B, Greenwood B, Kuile FOT. Post-discharge malaria chemoprevention in children admitted with severe anaemia in malaria-endemic settings in Africa: a systematic review and individual patient data meta-analysis of randomised controlled trials. *Lancet Glob Health*. 2024 Jan;12(1):e33-e44.

4. Ln81-82: Provide a reference for the statement

Reference added in the text.

Marwa K, Kapesa A, Baraka V, Konje E, Kidenya B, Mukonzo J, Kamugisha E, Swedberg G. Therapeutic efficacy of artemether-lumefantrine, artesunate-amodiaquine and dihydroartemisinin-piperazine in the treatment of uncomplicated *Plasmodium falciparum* malaria in Sub-Saharan Africa: A systematic review and meta-analysis. *PLoS One*. 2022 Mar 10;17(3):e0264339.

5. Ln83: Please make it clear that the low transmission setting you are referring to is not from Africa.

The text has been modified for better clarity, unambiguously informing the reader that we are referring to a SE Asian setting.

6. Ln85: “swiftly” is a relative term – so please provide some dates to add some context. The year the drug was introduced would help.

The text was modified, while the data of DHA-PPQ was nationwide implemented in Cambodia was added (2012), as well as a supporting reference [Chaorattanakawee et al., 2015]:

“...arose in multiple foci upon nation-wide introduction of DHA-PPQ in 2012, leading to the withdrawal of this artemisinin-based combination therapy (ACT) by the local authorities in 2016.” (page 4, lines 88-90)

Chaorattanakawee et al. Ex Vivo Drug Susceptibility Testing and Molecular Profiling of Clinical *Plasmodium falciparum* Isolates from Cambodia from 2008 to 2013 Suggest Emerging Piperazine Resistance. *Antimicrob Agents Chemother*. 2015 Aug;59(8):4631-43.

7. Ln87: Do you mean they occur in close proximity on chromosome 14?

Pfpm2 and 3 are part of a dense cluster of four plasmepsin genes (*pfpm1-4*) in a ca. 25 kb spanning region on chromosome 14. The text was changed for enhanced clarity:

“DHA-PPQ treatment failures have been linked to increased copy numbers of the *P. falciparum* plasmepsin 2 (*pfpm2*) and 3 (*pfpm3*) genes, part of a dense cluster of four plasmepsin genes (*pfpm1-4*), in a ca. 25 kb spanning region at chromosome 14” (page 4, lines 90-93)

8. Ln89: Insert “are” before “involved”

Thank you for pointing this out, this is corrected.

9. Ln93: Do mean less sensitive to ACTs in general or specifically piperaquine?

We are specifically referring to piperaquine. The text was modified accordingly:

“In African high transmission areas, slowly declining subtherapeutic concentrations of piperaquine after DHA-PPQ treatment provide ample opportunities for selecting re-infecting parasites less sensitive to piperaquine.” (page 5, lines 96-98).

10. Ln95: Do not understand the point being made here. If there is resistance, it should be an issue there the drug is first or second line as the treatment will still fail. Please rephrase for clarity.

The text was simplified to convey the message that the future might see an expansion in DHA-PPQ use, in response to the present trend of decreased artemether-lumefantrine efficacy:

“In addition, a future increase in DHA-PPQ usage in Africa is conceivable, should treatment failure rates with artemether-lumefantrine continues to increase on the continent” (page 5, lines 100-102).

11. Ln101: Please expand on what analysis was done. Were you assessing acceptability, efficacy, or cost-effectiveness?

We have reworded the text, but wish to point out that the focus of our analysis, expressed in the manuscript text (“...to investigate the durability of the post-treatment prophylactic efficacy of dihydroartemisinin-piperaquine and in parallel assess the evolution of *pfpm2* and *pfpm3* copy number variation.”):

The treatment efficacy, as well as the other aspects referred by the reviewer, as well as issues pertaining safety and efficacy (as per EMA regulatory demands, <https://www.mmv.org/newsroom/news-resources-search/pyramaxr-granules-becomes-first-paediatric-antimalarial-receive-ema>), have been provided by the original WANECAM Consortium report [WANECAM, 2018]. We refer the reviewer to this publication and the rather extensive linked supplementary material.

West African Network for Clinical Trials of Antimalarial Drugs (WANECAM). Pyronaridine-artesunate or dihydroartemisinin-piperaquine versus current first-line therapies for repeated treatment of uncomplicated malaria: a randomised, multicentre, open-label, longitudinal, controlled, phase 3b/4 trial. *Lancet*. 2018 Apr 7;391(10128):1378-1390.

12. Ln103: I assume you mean patients were followed up for 2 years with DOT for each repeat infection, rather than only patients with repeat infections were followed up for 2 years? Please clarify.

That is correct – each patient was part of the trial for a period of two years, during which every new malaria infection was managed with directly observed treatment and followed up for 63 days. If a new infection would happen inside the follow up, the patient was again treated with the same treatment according to the randomised

allocation to either DHA-PPQ or ASAQ for the first episode at enrolment. After 63 days, the protocol stipulated passive follow-up, i.e., patients or caretakers were asked to present to the study team in case of fever. If the cause of fever was attributable to malaria, the patient was also retreated in the same way as within the 63-day active follow-up window. Each treatment gave rise to a new post-treatment follow-up period, i.e., a 63-day active follow-up was succeeded – and in case, no re-infection was detected, by a passive follow-up. These cycles were repeated for a maximum of 2 years for each study participant. Please also note our response to comment #11 above.

We have amended the text to improve clarity:

“Here we present a retrospective clinical and molecular analysis of the dihydroartemisinin-piperaquine versus the artesunate-amodiaquine (ASAQ) arm, part of the multicentre WANECAM randomized controlled trial performed in Southern Mali, involving a passive surveillance period of two years per patient with repeated, directly observed treatment for each new malaria episode³. This unique design allowed us to investigate the durability of the post-treatment prophylactic efficacy of DHA-PPQ, while assessing the parallel evolution of *pfpm2* and *pfpm3* copy number variation as putative piperaquine resistance factors.” (page 5, lines 108-115).

Results:

1. Ln111: Please check if months need to be written out in full. Also, state that malaria was confirmed in these patients recruited to receive the specific treatment.

The text has been modified according to the suggestions:

“Between January 16, 2012, and May 18, 2013, 225 and 224 patients with microscopically confirmed uncomplicated malaria were recruited to receive DHA-PPQ and artesunate-amodiaquine, respectively.” (page 6, lines 120-122).

2. Ln114: Was the shortest time for follow-up in the DHA-PPQ arm 2 days? This person should have been excluded from the study given the short follow-up time.

This was an unfortunate typo; the minimum reinfection time was 16 days. The text was changed accordingly.

3. Ln115: Was the difference in repeat infections significantly different? If possible, provide a p value here.

What we report here are the actual number of reinfections in the respective study arms (providing a description of the study cohort), not averages or other summary statistics. Statistical comparisons are detailed in the “Rapid decline of piperaquine post-treatment protection efficacy” section.

4. Ln122: Were there any minor adverse events? If so, they should be listed.

As mentioned in a previous response, the issue of adverse events associated with the studied clinical trial is not part of the objectives of this report. This information is published in detail in the original report and its linked supplementary online materials [WANECAM, 2018]. Nevertheless, we note again that there were no occurrences of drug associated serious adverse events in the studied cohort.

West African Network for Clinical Trials of Antimalarial Drugs (WANECAM). Pyronaridine-artesunate or dihydroartemisinin-piperaquine versus current first-line therapies for repeated treatment of uncomplicated malaria: a randomised, multicentre, open-label, longitudinal, controlled, phase 3b/4 trial. *Lancet*. 2018 Apr 7;391(10128):1378-1390.

5. Ln125: This is a discussion point – just describe your result here, the findings can be explained in the discussion.

As mentioned in our response to the other reviewer, there are no guidelines against this type of presentation, and we wish to maintain this format unless advised against by the editor

6. Ln128: Was the difference in incidence rate significant between the two treatments?

Fig 1A shows the incidence density as a continuous variable and not a summary statistic for a particular date or time period that can be compared by standard statistical methods (i.e. null hypothesis testing and the determination of significance). However, the risk ratios shown in Figure 1B show that there was a statistically significant difference in risk (relative to ASAQ) at the start of the study, which then disappeared after around 12 months.

7. Ln136: Did the risk stay the same for ASAQ?

We have calculated the risk of re-infections in the dihydroartemisinin-piperaquine arm at different time points during the study *relative* to the risk of re-infections in the artesunate-amodiaquine arm, i.e., presenting an analysis of the relative risk. We consider this to be the appropriate way for analysing data from a randomised controlled trial, which takes care of a multitude of potential confounders – most importantly, changes in the transmission intensity over time.

8. Ln137: Do not understand the point being made. Please clarify. Also state which drug you are talking about here.

We have changed the text to improve clarity: “In accordance to the observed increase in the relative infection risk over time, the average time to re-infection in the dihydroartemisinin-piperaquine arm decreased significantly between 2012 and 2014 (P<0.005 ANOVA; 2012: mean 86 days, IQR [59, 107]; 2013: mean 72 days, IQR [51, 87]; 2014: mean 72 days, IQR [48, 76]) (Figure 1C).” (page 7, lines 149-153).

9. Ln145: Why is only plasmepsin 3 mentioned in the title when you investigated both plasmepsin 2 and plasmepsin 3?

Due to the rarity of *pfpm2* increased copy number, which prevented us from drawing definitive conclusions regarding its influence on the time to reinfection, we refer only to *pfpm3* in the title in order to highlight the central conclusion of our work. However, the title was slightly modified to better express this intention: “Decreased dihydroartemisinin-piperaquine protection against recurrent malaria associated with *Plasmodium falciparum* plasmepsin 3 copy number variation in Africa”

10. Ln154: Was the presence of 6 infections with increased plasmepsin 3 CNV significant? Were these repeat infections?

They were not significant, and they were all reinfections in different patient’s infections.

11. Ln158: Please quantify this increase over time – what was the number at baseline, after year one and at the end of the study.

Data was added in the text, as requested:

“We observed an increase frequency along time of mixed infections carrying *pfpm3* multi-copy variants (Figure 2A, B, mean normalized CNV scores 2012: 1.01, 95% CI [0.99,1.03]; 2013: 1.01, 95% CI [0.99, 1.03]; 2014: 1.17, 95% CI [1.11, 1.22], figure 2B)” (pages 7-8, lines 169-176)

12. Ln165: some quantification of the number of repeat samples with CNV needs to be provided in the text.

As mentioned in our response to a similar question by the previous reviewer, malaria infections in Africa are highly complex. (i.e., containing several clones), characterised by a massive genetic diversity which leaves us with the challenge of quantifying the number of infections carrying parasite subpopulations harbouring increased *pfpm3* copy numbers. As previously referred, the issue in the African high transmission areas is that sub-populations of parasites with more than 1 copy of *pfpm3* co-occur in multiclonal infections with wildtype, single-copy parasites. Therefore, the CNV score reported here is a population-level average which leads to the key observations of (a) the association between CNV and time-to-reinfection and (b) the increase in average CNV over the duration of the study.

13. Ln167: Do not understand the point being made. Please rephrase for clarity.

In SE Asia, *pfpm2* and *pfpm3* duplications linked to piperazine resistance result from the formation of a full *pfpm2* duplication but also the formation of a *pfpm1-3* hybrid [Amato et al., 2017], which is responsible for the *pfpm3* 2-copy signal. This is what we refer to as linked. This 1-3 hybrid was not found in our study in accordance with a specific inverse PCR method for the detection of the hybrid breakpoint [Ansbro et al., 2020]. We recognize the text as confusing, and even including a mistake. The text was changed in order to clarify this issue:

“*pfpm2* and *pfpm3* CNV events were not found to be present in the same infection. We also did not find evidence for the presence of the structural breakpoint associated to the *pfpm1-3* hybrid found in South East Asia^{7,14}.”

No statistically significant signal was found for the positive selection by piperazine of any of the tested *pfprt* SNPs during the follow up.”

Amato R, Lim P, Miotto O, Amaratunga C, Dek D, Pearson RD, Almagro-Garcia J, Neal AT, Sreng S, Suon S, Drury E, Jyothi D, Stalker J, Kwiatkowski DP, Fairhurst RM. Genetic markers associated with dihydroartemisinin-piperazine failure in *Plasmodium falciparum* malaria in Cambodia: a genotype-phenotype association study. *Lancet Infect Dis.* 2017 Feb;17(2):164-173. doi: 10.1016/S1473-3099(16)30409-1. Epub 2016 Nov 3. PMID: 27818095; PMCID: PMC5564489.

Ansbro MR, Jacob CG, Amato R, Kekre M, Amaratunga C, Sreng S, Suon S, Miotto O, Fairhurst RM, Wellems TE, Kwiatkowski DP. Development of copy number assays for detection and surveillance of piperazine resistance associated plasmepsin 2/3 copy number variation in *Plasmodium falciparum*. *Malar J.* 2020 May 13;19(1):181. doi: 10.1186/s12936-020-03249-x. PMID: 32404110; PMCID: PMC7218657.

14. Ln170: Are you saying the *pfprt* SNPs were equally present in both study arms or that no SNP was selected for in the DHA-PPQ arm? Please clarify.

We apologise for the confusion. What we meant to say is that we did not observe a significant difference in *pfprt* SNP frequency between baseline infections (prior to first treatment) vs recurrent infections in the DHA-PPQ arm; *pfprt* SNPs were not evaluated in the ASAQ arm since we focused on identifying molecular correlates for our epidemiological finding of a substantial reduction of the post-treatment prophylactic efficacy DHA-PPQ. The text was changed for improved clarity.

15. Ln171: Was this association in re-infection time was present in both study arms or only the DHA-PPQ arm?

As mentioned above, *pfprt* SNPs were solely analysed for the DHA-PPQ arm as the post-treatment protective effect of this combination was the focus of our study.

Discussion:

1. Ln180-181: This statement makes it sound like the patients got a different drug following the first infection and only if they returned were they treated with either DHA-PPQ or ASAQ. Please clarify.

We have modified the sentence for better clarity:

“Here we have analysed a unique large clinical trial dataset with 1,430 *P. falciparum* malaria episodes in 449 patients randomized to receive either DHA-PPQ or artesunate-amodiaquine for the first episode at recruitment, as well as for each subsequent *P. falciparum* recurrence over an extended follow-up of 2 years in a high transmission setting in Africa”. (page 8, 195-199)

2. Ln182: I thought more follow-ups are up to 28 days – please check.

Although the WHO has long recommended 42-day follow-up periods for therapeutic efficacy studies involving DHA-PPQ (WHO, 2010), the reviewer is right in that a large number of studies include only a limited in 28 days follow-up, as recently shown in a meta-analysis by Marwa et al., 2022. The text was changed accordingly. “...efficacy studies characterized by shorter active follow-ups, typically up to day 28.” (pages 8-9, lines 200-203).

WHO Global Malaria Programme. *Guidelines for the treatment of malaria*. Geneva: World Health Organization, 2010.

Marwa K, Kapesa A, Baraka V, Konje E, Kidenya B, Mukonzo J, Kamugisha E, Swedberg G. Therapeutic efficacy of artemether-lumefantrine, artesunate-amodiaquine and dihydroartemisinin-piperaquine in the treatment of uncomplicated *Plasmodium falciparum* malaria in Sub-Saharan Africa: A systematic review and meta-analysis. *PLoS One*. 2022 Mar 10;17(3):e0264339.

3. Ln184: This is a moot point as most studies are not designed to assess prolonged prophylactic effects.

We agree with the critique and modified the text accordingly:

“This trial markedly differs from the majority of antimalarial, single-episode, efficacy studies characterized by shorter active follow-ups, typically up to day 28¹⁴, not designed to assess the post-treatment prophylactic efficacy over long periods.” (pages 8-9, lines 199-203).

4. Ln187-188: Since drugs start off highly effective, with efficacy declining as resistance mutations are acquired, I am assuming you mean this study allowed you to investigate the evolution of resistance/reduced efficacy. Please clarify.

This is indeed the key message of our work (though again to clarify, we are not reporting a decline of therapeutic efficacy but are studying the post-treatment prophylactic efficacy). Our study design is key for understanding the evolution of resistance, especially during the period when the changes in the chemosensitivity profile of parasites are still clinically “silent”. In other words, whilst such small chemosensitivity changes can provide a competitive advantage over wildtype parasites and give rise to earlier reinfections (when exposed to trailing subtherapeutic drug concentrations after treatment) they are not (yet) permitting survival during treatment characterised by peak drug concentrations, thus not affecting the therapeutic efficacy of a combination therapy. Additionally, we would like to re-emphasise the fact that the value of long-acting/slowly eliminated artemisinin combination partner drugs such as piperaquine goes beyond the acute treatment phase. The slow elimination is critical for post-treatment prophylactic efficacy, critical for their use in the expanding portfolio of chemo preventive strategies like seasonal malaria chemoprevention (SMC), intermittent preventive treatment in pregnancy (IPTp), perennial malaria chemoprevention (PMC) or post-discharge malaria chemoprevention (PDMC).

5. Ln191-192: I assume you mean provide longer protection against reinfection. Please clarify.

This is correct; we have changed the text accordingly:

“...DHA-PPQ is being recommended and advocated on its capacity to provide longer protection against reinfections.” (page 9, lines 207-208).

7. Ln202: Do not understand what “complex combinations” means. Please clarify.

We are referring to the relatively tortuous development of piperazine combinations, including China-Vietnam 8 (CV8) and Artecom [Davis et al., 2005]. The phrasing was simplified. (Page 9, lines 216-219).

Davis TM, Hung TY, Sim IK, Karunajeewa HA, Ilett KF. Piperazine: a resurgent antimalarial drug. *Drugs*. 2005;65(1):75-87. doi: 10.2165/00003495-200565010-00004. PMID: 15610051.

8. Ln218: Did they carry CNV in both the plasmepsin 2 and 3 genes?

No, these amplifications were present in different infections, so what we refer to is the set carrying *pfpm2* or *pfpm3*, not both simultaneously, according to the 1.5 CNV score threshold. We understand that the agglomerated notation leads to obvious confusion. We have changed the text as recommended.

9. Ln219-220: Do not understand the point being made. Please rephrase for clarity.

We refer to the fact that both *pfpm2* and *pfpm3* increased copy number, in accordance to the 1.5 CNV score threshold, were not detected in the baseline infections (i.e. at trial recruitment point, corresponding to the first episode). We interpret that such mutant parasites detected among the follow-up reinfections resulted from piperazine-driven selection by the drug's post-treatment pharmacokinetic tail.

We have rephrased the text to simplify this point: "samples from infections with short periods between treatment and subsequent episodes were found enriched in *pfpm2* and *pfpm3* CNV carrying parasites." (page 10, lines 238-239).

10. Ln224: I am assuming you mean reinfection infections with increased copy numbers were more likely to be detected follow DHA-PPQ treatment? Please clarify.

Yes. Such infections are more likely to be detected because they are expected to be able to re-infect a subject with trailing subtherapeutic piperazine concentrations. Essentially, they are more tolerant to the drug which confers an advantage of these population of parasites when compared with fully wildtype infections. This observation is reminiscent of the original observation of the post/treatment selection of *pfmdr1* N86 allele upon artemether-Lumefantrine treatment [Sisowath et al., 2005; Hastings and Ward, 2005].

Sisowath C, Strömberg J, Mårtensson A, Msellem M, Obondo C, Björkman A, Gil JP. In vivo selection of *Plasmodium falciparum* *pfmdr1* 86N coding alleles by artemether-lumefantrine (Coartem). *J Infect Dis*. 2005 Mar 15;191(6):1014-7. doi: 10.1086/427997. Epub 2005 Feb 8. PMID: 15717281.

Hastings IM, Ward SA. Coartem (artemether-lumefantrine) in Africa: the beginning of the end? *J Infect Dis*. 2005 Oct 1;192(7):1303-4; author reply 1304-5. doi: 10.1086/432554. PMID: 16136476.

11. Ln239-240: Do not understand the point being made. Please rephrase for clarity.

Presently, *pfkelch13* mutations associated with partial artemisinin resistance are emerging and spreading mostly in the East African region [Rosenthal et al., 2024]. In accordance with the experience in SE Asia [Phyo et al., 2016], this in turn drives the progressive erosion of the partner drug performance, leading to the ultimate failure of the ACT as a whole.

The text was slightly modified for clarity:

"could potentiate the risk of treatment failures in case proper partner drug protection is not preserved" (page 11, lines 256-257).

Phyo AP, Ashley EA, Anderson TJC, Bozdech Z, Carrara VI, Sriprawat K, Nair S, White MM, Dziekan J, Ling C, Proux S, Konghahong K, Jeeyapant A, Woodrow CJ, Imwong M, McGready R, Lwin KM, Day NPJ, White NJ, Nosten F. Declining Efficacy of Artemisinin Combination Therapy Against *P. falciparum* Malaria on the Thai-Myanmar Border (2003-2013): The Role of Parasite Genetic Factors. *Clin Infect Dis*. 2016 Sep 15;63(6):784-791. doi: 10.1093/cid/ciw388. Epub 2016 Jun 16. PMID: 27313266; PMCID: PMC4996140.

12. Ln242: Why only plasmepsin 3?

The reviewer is correct. As *pfpm2* is clearly selected among recurrent parasitemias, we reinforce the recommendation for the analysis of both plasmepsin 2 and 3 genes. The text was modified accordingly: “Regular determination of *pfpm2* and *pfmp3* average CNV...” (page 11, lines 259-260)

Methods:

1. Ln274: Suggestion ethics approval numbers are reported here.

This study was registered at the Pan African Clinical Trials Registry, number PACTR201105000286876. The parasite genetic analysis was approved by the Ethical Committee of the Faculte de Medecin et D’Odonto-Stomatology, ref. 2010_79_FMPOS as well as by Stockholm Regional Ethics Committee, ref. 2016/2286-32 and 2017/499-32.

The information was added to the manuscript text (page 12, lines 290-294).

2. Ln276: So patients with any malaria species were enrolled into the study despite DHA-PPQ and ASAQ recommended for falciparum malaria? Please confirm.

Infections carrying *P. falciparum* were included independently of the presence of other *Plasmodia* species. The occurrence of these parallel infections was nevertheless documented [WANECAM, 2018].

3. Ln287-288: Please provide more information on how and what done during passive and active follow-up.

We would like to refer to our original publication [WANECAM, 2018] for a detailed description of the trial, as well as the information available in its registration (Pan African Clinical Trials Registry (PACTR201105000286876). We added a more informative text:

“After inclusion and after each subsequent episode patients were followed up actively (weekly by microscopy) and after 63 days, passively by inviting study participants to present for any subsequent fever episode for a period of two years.” (page 13, lines 305-308).

4. Ln290: Information on msp1 and 2 marking was not presented in this paper. So were all repeat infections new infections?

Virtually all recurrences were classified as reinfections in accordance to *pfmsp1/2* + *glurp* analysis, a fact which was reflected by the very high efficacy reported for both DHA-PPQ and ASAQ (>98%). Further details are provided by the original publication [WANECAM, 2018]. As stated above, for the specific purpose of studying the selection of *pfpm2* or 3 copy number variation by trailing plasma concentrations of piperaquine after treatment with dihydroartemisinin-piperaquine, a distinction between recrudescence and new infections does not add further information since both types of infections are considered to be under selection pressure.

5. Ln310: provide a reference for the *pfprt* SNP marking.

The SNPing *pfprt* position 326 and 356 were assessed with PCR-RFLPs in-house developed for this work; details on the methods are provided in the supplementary information.

6. Ln320: Why only plasmepsin 3?

The rarity of *pfpm2* increased copy number at baseline, associated with the lack of evidence for *pfpm1/3* hybrid genes, as well as the reported non-importance of such mutations in the parasite response to amodiaquine mutation [Mairet-Khedim et al., 2021], led us to conclude that the analysis of *pfpm2* for the amodiaquine arm would not provide us with significant information.

Mairet-Khedim M, Leang R, Marmai C, Khim N, Kim S, Ke S, Kaoy C, Kloeung N, Eam R, Chy S, Izac B, Mey Bouth D, Dorina Bustos M, Ringwald P, Arieu F, Witkowski B. Clinical and In Vitro Resistance of Plasmodium falciparum to Artesunate-Amodiaquine in Cambodia. Clin Infect Dis. 2021 Aug 2;73(3):406-413. doi: 10.1093/cid/ciaa628. PMID: 32459308; PMCID: PMC8326543.

7. Ln332: Do not understand why risk ratios could not be calculated for samples without filter paper samples. Please more information for clarity.

We apologise for the confusion, motivated by a typo - the issue was not the lack of filter paper samples but any samples in the DHA-piperaquine arm in 2015, when this study came to an end.

This has been corrected in the text:

“For this analysis we only considered the period 2012-2014 due to the lack of samples from 2015.” (page 15, lines 354-355).

8. Ln333: Are you suggesting that there is no malaria transmission during the low season? Please provide a reference to support this statement.

We do not claim that malaria transmission totally stops, but it decreases significantly, as shown in Figure 1A, which is also the basis of the implementation of WHO-recommended seasonal malaria chemoprevention (SMC) in the region. This sharp decrease implicates that reinfection times can be very distorted due to low transmission rates during the off season as exposure / infection risk is much lower. It becomes a confounding effect when evaluating the post-treatment protection effect of piperaquine as a patient treated towards the end of the season will appear to benefit in its post-treatment period due to the natural decrease in re-infection risk. For that reason, we kept our analysis inside the specified time window.

A contemporary reference [Coulibaly et al., 2013] was added, as advised.

Coulibaly D, Rebaudet S, Travassos M, Tolo Y, Laurens M, Kone AK, Traore K, Guindo A, Diarra I, Niangaly A, Daou M, Dembele A, Sissoko M, Kouriba B, Dessay N, Gaudart J, Piarroux R, Thera MA, Plowe CV, Doumbo OK. Spatio-temporal analysis of malaria within a transmission season in Bandiagara, Mali. Malar J. 2013 Mar 1;12:82. doi: 10.1186/1475-2875-12-82. PMID: 23452561; PMCID: PMC3618208.

Reviewer #1 (Remarks to the Author):

Thank you for your response to the reviews and your helpful revision. I am still not clear which differences are or are not statistically significant because figures show trends without statistical testing results and results in the text generally do not show statistical significance. I think if these can be clarified, the manuscript has important information to inform surveillance for drug resistance moving forward.

Firstly, we are grateful for reviewer 1 for his/her further effort to help improving our manuscript.

Concerning the specific request, we have included more information on significant differences in the legends of figures 1C and 2B [lines 555-571], highlighting the importance of *pfpm3* increased copy number in the parasite response to piperazine, and the associated selection of the marker along the period of the trial. Figure 1 legend was also altered for clarification [figure 2, lines 585-588].

Consequence of the added information, a number of informative changes in the text were introduced on some sections of the text, for for keeping the manuscript as streamlined and clear as possible. In section “Rapid decline of piperazine post-treatment protection efficacy” changes at lines 142-171, 173, 176 and 179. Small changes also on the “Post-treatment selection of plasmepsin 3 gene copy number variation following DHA-PPQ treatment” section, lines 198-203. In the “Discussion” section, lines 246-248

Also, and considering the recent publication of the 2024 World Malaria Report, more up-to-date data was included in the introduction [lines 67-68]. Updates in the bibliography were also added.